# Cellular and Molecular Targets of Nucleotide-Tagged Trithiolato-Bridged Arene Ruthenium Complexes in the Protozoan Parasites *Toxoplasma gondii* and *Trypanosoma brucei*

**DOI:** 10.3390/ijms221910787

**Published:** 2021-10-05

**Authors:** Nicoleta Anghel, Joachim Müller, Mauro Serricchio, Jennifer Jelk, Peter Bütikofer, Ghalia Boubaker, Dennis Imhof, Jessica Ramseier, Oksana Desiatkina, Emilia Păunescu, Sophie Braga-Lagache, Manfred Heller, Julien Furrer, Andrew Hemphill

**Affiliations:** 1Institute of Parasitology, Vetsuisse Faculty, University of Bern, 3012 Bern, Switzerland; nicoleta.anghel@vetsuisse.unibe.ch (N.A.); ghalia.boubaker@vetsuisse.unibe.ch (G.B.); dennis.imhof@vetsuisse.unibe.ch (D.I.); jessi95@gawnet.ch (J.R.); 2Institute of Biochemistry and Molecular Medicine, University of Bern, 3012 Bern, Switzerland; mauro.serricchio@ibmm.unibe.ch (M.S.); jennifer.jelk@ibmm.unibe.ch (J.J.); peter.buetikofer@ibmm.unibe.ch (P.B.); 3Department of Chemistry, Biochemistry and Pharmaceutical Sciences, University of Bern, 3012 Bern, Switzerland; oksana.desiatkina@unibe.ch (O.D.); paunescu_emilia@yahoo.com (E.P.); julien.furrer@unibe.ch (J.F.); 4Proteomics and Mass Spectrometry Core Facility, Department for BioMedical Research (DBMR), University of Bern, 3012 Bern, Switzerland; sophie.lagache@dbmr.unibe.ch (S.B.-L.); manfred.heller@dbmr.unibe.ch (M.H.)

**Keywords:** mitochondrion, affinity chromatography, binding proteins, ATP-synthase, proteomics, metabolism

## Abstract

*Toxoplasma gondii* is an apicomplexan parasite that infects and proliferates within many different types of host cells and infects virtually all warm-blooded animals and humans. *Trypanosoma brucei* is an extracellular kinetoplastid that causes human African trypanosomiasis and Nagana disease in cattle, primarily in rural sub-Saharan Africa. Current treatments against both parasites have limitations, e.g., suboptimal efficacy and adverse side effects. Here, we investigate the potential cellular and molecular targets of a trithiolato-bridged arene ruthenium complex conjugated to 9-(2-hydroxyethyl)-adenine (**1**), which inhibits both parasites with IC_50_s below 10^−7^ M. Proteins that bind to **1** were identified using differential affinity chromatography (DAC) followed by shotgun-mass spectrometry. A trithiolato-bridged ruthenium complex decorated with hypoxanthine (**2**) and 2-hydroxyethyl-adenine (**3**) were included as controls. Transmission electron microscopy (TEM) revealed distinct ultrastructural modifications in the mitochondrion induced by (**1**) but not by (**2**) and (**3**) in both species. DAC revealed 128 proteins in *T. gondii* and 46 proteins in *T. brucei* specifically binding to **1** but not **2** or **3**. In *T. gondii*, the most abundant was a protein with unknown function annotated as YOU2. This protein is a homolog to the human mitochondrial inner membrane translocase subunit Tim10. In *T. brucei*, the most abundant proteins binding specifically to **1** were mitochondrial ATP-synthase subunits. Exposure of *T. brucei* bloodstream forms to **1** resulted in rapid breakdown of the ATP-synthase complex. Moreover, both datasets contained proteins involved in key steps of metabolism and nucleic acid binding proteins.

## 1. Introduction

*Toxoplasma gondii* and *Trypanosoma brucei* belong to two evolutionary distant eukaryotic phyla, namely the clade Alveolata of the super-group Diaphoretickes in the case of *T. gondii*, and the group Euglenozoa of super-group Excavata in the case of *T. brucei* [1]. *T. gondii* causes important diseases in farm animals, has an enormous global economic impact, and has a high zoonotic potential [2]. In immunocompetent hosts, infection does not have serious consequences, and the proliferative tachyzoite form differentiates into the tissue cyst-forming bradyzoite stage, which can persist over many years to lifelong without causing any clinical symptoms. However, *Toxoplasma* is a major abortion-causing pathogen in sheep and other farm animals, and primary infection in pregnant women with *T. gondii* can lead to vertical transmission, causing fetal malformations and/or abortion [3]. In patients undergoing immunosuppressive disease or therapy, reactivation of bradyzoites from tissue cysts and re-differentiation into tachyzoites often causes serious pathology. Current drug treatments for toxoplasmosis therapy typically include antifolates using a combination of pyrimethamine–sulfadiazine or trimethoprim–sulfamethoxazole, and pyrimethamine can also be combined with clindamycin, azithromycin, or atovaquone. These treatments are unspecific, adverse effects have been frequently documented, and clinical failures have been reported [4]. While toxoplasmosis is a food-borne disease, human African trypanosomiasis (HAT) is a vector-borne disease, and transmission of *T. brucei gambiense* and *T. b. rhodesiense* to humans occurs via the tsetse fly [5]. Another *T. brucei* subspecies, *T. b. brucei*, is one of the causative agents of a wasting disease in cattle, called Nagana, which inflicts high economic and social impact in African rural areas [6]. Treatment options of HAT include pentamidine and suramin for the early and peripheral stage of the disease, and melarsoprol and eflornithine alone or in combination with nifurtimox for the cerebral phase [7]. More recently, 2-5-nitroimidazole fexinidazole has been approved for human use (https://dndi.org/research-development/portfolio/fexinidazole/; accessed on 15 June 2021). Veterinary trypanocides to treat Nagana include diminazene aceturate and isometam, and they are prone to drug resistance development. 

*T. gondii* and *T. brucei* occupy two different ecological niches. *T. gondii* is an intracellular parasite that resides and proliferates within a parasitophorous vacuole that is located within the cytoplasm of a host cell [8,9]. In contrast, *T. brucei* is an extracellular parasite, with the mammalian bloodstream forms undergoing antigenic variation and surviving in the blood and lymphatic vessels prior to crossing the blood–brain barrier, where they also remain in the extracellular space [10,11]. A common feature of both species is the presence of a single mitochondrion, which adopts a tube-like structure. The *T. gondii* mitochondrion typically exhibits an electron-dense matrix with clearly visible cristae structures, and components of the electron transport chain mediating oxidative phosphorylation are being regarded as highly valuable drug targets. In *T. brucei* bloodstream forms, the mitochondrion is largely devoid of discernible cristae, and the mitochondrial DNA is organized into a condensed structure named kinetoplast that is localized in a dilated area and is physically associated with the flagellar base and the flagellar pocket. Although the respiratory chain is not functional in *T. brucei* bloodstream forms, the mitochondrion is essential for parasite survival. Thus, compounds targeting mitochondrial function and structural integrity represent attractive drug candidates for the treatment of both parasitic infections.

In the last decade, extensive research has been carried out to identify suitable molecular drug targets in protozoan parasites that cause diseases in animals and humans [12,13]. Previously published in vitro studies have revealed that trithiolato-bridged dinuclear ruthenium(II)–arene complexes were effective against *T. gondii* [14,15,16,17], *Neospora caninum* [17,18], and *T. brucei* [19]. To improve the effectivity of these ruthenium complexes, their organic substituents have attracted special attention [20]. Since many parasites lack de novo synthesis pathways of essential metabolites and have therefore become auxotrophic [21], it is tempting to conjugate Ru(II) complexes with such metabolites. As these parasites need to scavenge essential components from their hosts, this could improve their uptake and binding to specific targets [22]. Trying to exploit purine auxotrophy [23], we have designed various conjugates based on a trithiolato-bridged dinuclear ruthenium(II)–arene scaffold with anchored purines or purine analogues [Desiatkina et al., 2021, manuscript in preparation].

In this study, we investigated the activity of compound **1**, which was identified in the screening for anti-*T. gondii* activity, containing a 9-(2-hydroxyethyl)-adenine group, against *T. brucei*. Compound **2**, a structurally related complex with 2-thioxanthine as one of the bridge thiols, and compound **3** (with 9-(2-hydroxyethyl)-adenine as one of the bridge thiols) were also assessed for comparison. Compounds **1–3** are shown in Figure 1. We also investigated whether common (sub-)cellular and/or molecular targets exist in these evolutionary and ecologically distant protozoan parasites using electron microscopy of cells treated with all three compounds. Previous studies on pull-downs with a trithiolato-bridged arene Ru(II) complex decorated with different substituents have suggested nucleic acid binding proteins, in particular elongation factor 1α and ribosomal proteins, as potential binding proteins [14]. However, the identification of these proteins was based on in-gel-digestion of protein bands after SDS-PAGE and was therefore biased.

## 2. Results

### 2.1. Compound ***1*** Inhibits Parasite Proliferation and Targets the Parasite Mitochondrion

Drug effects against T. gondii tachyzoites were assessed in vitro using a transgenic T. gondii strain grown in human foreskin fibroblast (HFF) monolayers that constitutively expresses β-galactosidase. Effects on *T. brucei* viability were determined by AlamarBlue® assay. Compound **1** inhibited the proliferation of T. gondii tachyzoites with an IC_50_ of 59 nM. but neither 2-thioxanthine compound **2** nor adenine derivative (**3**) exhibited notable activities (see Table 1; (Desiatkina et al., 2021, manuscript in preparation). Similar results were obtained with *T. brucei* bloodstream forms where **1** severely impaired parasite viability (IC_50_ = 29 nM), while the other two compounds remained largely ineffective. None of the compounds had any effect on HFF viability or morphology at a concentration of 2.5 µM (Desiatkina et al., 2021, manuscript in preparation).

Transmission electron microscopy (TEM) was performed to study the ultrastructural alteration during these treatments. Results for tachyzoites of the *T. gondii* Me49 strain are shown in Figure 2 and Figure 3. In non-treated control cultures (Figure 2A,B), tachyzoites were found within the cytoplasm of host cells, where they formed a parasitophorous vacuole that was delineated by a parasitophorous vacuole membrane. The apical complex with the conoid, and secretory organelles such as micronemes (mic), rhoptries (rop), and dense granules (dg) were clearly discernible. Tachyzoites harbor a single mitochondrion that displays an electron-dense and highly structured matrix, of which one or more portions are readily visible, depending on the section plane (Figure 2B). Upon treatment of cultures with 500 nM of **1**, the most notable changes occurred already after 6 h in the mitochondrial matrix, resulting in a complete disappearance of cristae and associated structures. However, the overall shape and structure as well as the secretory organelles of the parasites remained unaffected. In addition, the structural integrity of the parasitophorous vacuole was maintained, even after 12 h (Figure 3A) and 24 h of treatment (Figure 3B–D), with alterations in the mitochondria still evident. Occasionally, lipid droplets could be detected in the cytoplasm of treated parasites (Figure 3A). In some instances, tachyzoites appeared to be embedded in the matrix of the parasitophorous vacuole that had a rather solid structure; in other cases, the matrix appeared as a more loose and less electron-dense structural entity. No structural changes were evident upon treatment of *T. gondii* tachyzoites grown in host cells in the presence of **2** and **3** (data not shown).

TEM micrographs of *T. brucei* TREU927 bloodstream forms are shown in Figure 4. No structural differences were evident between non-treated bloodstream forms and trypanosomes treated with **2** or **3.** Parasites displayed the typical hallmarks of trypanosomatids. These include the flagellum (f) with an axoneme and the paraflagellar rod, which emerges from the posterior end and then runs along the entire body toward the anterior part of the cell. In addition, portions of the mitochondrion (mito) were visible, depending on the section plane, surrounded by a double membrane and filled with a fine, slightly electron-dense matrix.

Cristae or cristae-like structures were occasionally protruding from the inner membrane into the mitochondrial matrix (Figure 4A–D). The membrane stacks of the Golgi apparatus appeared normal (Figure 4D, and the kinetoplast (k), located at the posterior end, and also integrated into the mitochondrion, was identifiable as a highly ordered entity (Figure 4E).

In contrast, *T. brucei* bloodstream forms treated with 200 nM of **1** for 4 h showed distinct early ultrastructural alterations, most notably in the mitochondrion (Figure 5). Compound **1** induced swelling of the mitochondrion. The matrix appeared less electron-dense compared to the controls and was devoid of cristae-like membranous protrusions but often contained non-defined filamentous or membranous components. In many instances, the kinetoplast had lost its characteristic overall electron-dense appearance and was fragmented (Figure 5B,C). In addition, in some instances, parasites were observed that had undergone partial leakage of cytoplasmic content, which was reflected by a lower electron density of the cytoplasm, and a deterioration of structural organization of the cells, as seen by the presence of flagellar structures within the cytoplasm (Figure 5F).

### 2.2. Identification of Compound-Binding Proteins in T. gondii and T. brucei

To identify the molecular targets of these compounds, the pull-down fractions obtained from *T. gondii* ME49 tachyzoites were identified by differential shotgun mass spectrometry. Overall, 2542 unique peptides matching to 375 proteins were identified (Appendix A). In the pull-down fractions obtained from *T. brucei* TREU927 blood stream forms, 975 unique peptides matching to 198 proteins were identified (Appendix A). Overall analysis of both datasets by principal component analysis using LFQ protein intensity values revealed that the protein patterns obtained from mock and compound **3**-colums clustered together. The protein patterns obtained from **1** and **2** columns were separated by the first and second principal component from each other and from the **3** and mock columns (Figure 6).

A closer look at the *T. gondii* pulldown revealed that of the 277 proteins that did not bind to the mock column, 122 proteins were specifically retained by compound **1**. A total of 130 proteins retained by **1** were also identified in the column **2** and/or **3** fractions. In the *T. brucei* pulldown, 44 of the 94 proteins not retained by the mock column were specifically eluted from the compound **1** column, and 34 proteins were also eluted from the other columns (Figure 7).

### 2.3. The Major Compound ***1***-Binding Proteins

Amongst the 122 *T. gondii* proteins, which were specifically retained by **1**, the protein encoded by the open reading frame (ORF) TGME49_319730, annotated as a YOU2 family C2C2 zinc finger protein, was—by far—the most abundant one with ca. 30% of the binding proteins (Table 2).

However, we were unable to identify the C2H2 zinc finger motif x(2)-Cys-x(2,4)-Cys-x(12)-His-x(3,5)-His [24] in the primary sequence of the deduced protein (Figure 8A). Homologs to the polypeptide encoded by ORF TGME49_319730 were identified in other apicomplexans, fungi and vertebrates. Alignments with the conserved hypothetical protein NCLIV_010410 from *Neospora caninum* (87% identity), the protein FAM136A from *Cyclospora cayetanensis* (XP_022591373.1; 46% identity), the conserved protein XP_001349642.1 from *Plasmodium falciparum* 3D7 (33% identity), the protein FAM136A from *Clupea harengus* (XP_012669662.1; 25% identity), and the human mitochondrial import inner membrane translocase subunit Tim10 (NP_036588.1; 12.5% identity) are depicted in Figure 8B. To identify a possible function of this protein, a modeling was performed by comparing a potential structure to existing structures sharing similarities. In the Swiss Model repository, the template with the highest identity was 7cgp.1.O, the mitochondrial import membrane translocase subunit Tim10 encoded by NP_036588.1, which forms a 15-mer, as published in a recent cryo-EM study [25]. The model for ORF TgME49_319730 with a convenient QMEAN value (−0.47) is shown in Figure 8C.

The second most abundant protein was TGME49_316710 encoding a hypothetical protein. Moreover, a class I fructose-1,6-bisphosphate aldolase [26] was amongst the most abundant compound **1**-binding proteins (Table 2; see Appendix A for complete list).

The results obtained with the *T. brucei* pull-down were even sharper. Amongst the 48 proteins, which were specifically retained by compound **1**, more than 64% were constituted by five mitochondrial ATP synthase subunits, the subunit encoded by ORF Tb927.5.3090 alone amounting to nearly 25% of the specific compound **1**-binding proteins (Table 3; see Appendix A for complete list).

In *T. gondii*, no mitochondrial, but vacuolar ATP-synthase subunits, which are homologs, were among the specific proteins binding to **1**. Furthermore, in both pulldowns, proteins involved in translation such as initiation and elongation factors and acyl-transferases were specifically eluted from compound **1** columns (Table 1 and Table 2; Appendix A).

### 2.4. Compound **1** Impairs the Stability of the T. brucei ATP-Synthase Complex

The *T. brucei* ATP-synthase complex is composed of two subcomplexes, the membrane-embedded F_o_ and the soluble F_1_, which are connected by a rotary central stalk and a stationary peripheral stalk [27]. The effect of drug treatment on the stability of the *T. brucei* ATP-synthase complex was investigated by exposing bloodstream forms to 50–400 nM compound **1** or **2** for 1 h, and extracts were separated by native PAGE (Figure 9A). Treatment with 200 nM or 400 nM compound **1** resulted in a significantly decreased amount of the ATP-synthase complex, while treatment with compound **2** had no effect (Figure 9B). In contrast, another mitochondrial complex, the ADP/ATP carrier protein (TbAAC), which is responsible for the exchange of free ADP and ATP across the inner mitochondrial membrane [28], was not affected by these treatments.

## 3. Discussion

We here report on a trithiolato-bridged dinuclear Ru(II)–arene compound **1** that is conjugated to a 9-(2-oxyethyl)-adenine unit, which is highly active against two evolutionary distinct protozoans, namely *T. gondii* belonging to the super-group Diaphoretickes (clade Alveolata) and *T. brucei* belonging to the group Euglenozoa of the super-group Excavata [1]. Another di-ruthenium compound containing 2-thioxantine as one of the thiolato bridges **2** and 9-(2-hydroxyethyl)-adenine **3** displayed no in vitro activity against these two species. Thus, we were interested in the cellular and molecular targets in *T. gondii* and *T. brucei* with which compound **1** would specifically interact.

Cellular targets were investigated by TEM. In both parasites, compound **1** induced severe ultrastructural alterations most notably in the mitochondrion already 4–6 h after exposure to the drug, such as the swelling and dissolution of cristae and the mitochondrial matrix. In addition, in many instances, the matrix was replaced by membranous or filamentous material of unknown nature. In *T. brucei,* treatment with the drug also affected the structural integrity of the kinetoplast, which contains the mitochondrial DNA. While TEM suggested that other organelles were largely unaffected, it is important to note that the absence of ultrastructural evidence does not exclude the possibility that also other cellular entities might be affected by the drug.

These findings are in line with previous studies on a series of thiolato-bridged dinuclear ruthenium(II)–arene complexes that induced similar effects in both *T. gondii* and the closely related *Neospora caninum* [16,18], and also *T. brucei* bloodstream forms [19]. In the case of *T. brucei*, this type of compound also severely impaired the mitochondrial membrane potential.

When examining the overall pull-down results, it is striking that **1** has more common binding proteins with **2** than with its appended moiety 9-(2-oxyethyl)-adenine (3) (see Figure 1), which had only few binding proteins, most of them in common with the other compounds. This observation suggests a general high affinity of proteins to the cationic trithiolato di-ruthenium core structure of both compounds. It is worth noting also that both **1** and **2** contain a purine nucleobase in the structure. However, if in the case of **1** adenine is separated from the di-ruthenium unit by the presence of a short spacer, in **2**, 2-thioxantine represents one of the bridges connecting the two ruthenium(II)-arene fragments. Then, these structural differences would trigger the specificity of the binding.

Only few proteins from both organisms binding to compound 1 columns are homologs, namely translation initiation and elongation factors and ribosomal proteins. This is in good agreement with previous findings obtained previously with another trithiolato-bridged ruthenium–arene complex [14]. The translation machinery as a common target in both organisms is further evidenced when examining the protein–protein interaction network generated by the STRING knowledge base and software tool (Swiss Institute of Bioinformatics) for the compound **1**-binding proteins from *T. gondii* (Appendix A) and *T. brucei* (Appendix A). However, protein networks show differences due to the different numbers of binding proteins identified in both organisms. Other proteins share common metabolic specifications such as nucleotide binding or transferase activities.

In both *T. gondii* and *T. brucei*, the most abundant proteins specifically binding to **1** are—in all likelihood—located to the mitochondrion. This is in a good agreement with the subcellular damage seen by TEM. In *T. brucei*, the observed destabilization of the major binding protein—the mitochondrial ATP synthase complex—would seriously affect mitochondrial integrity. In *T. brucei*, the role of the ATP synthase depends on the life cycle stage. In the insect/procyclic stage, the ATP synthase complex is responsible for producing ATP, while it consumes ATP in bloodstream form parasites. This switch is accompanied by changes in the mitochondrial ultrastructure. In addition to other functions, the mitochondrial ATP synthase is responsible for maintaining the mitochondrial membrane potential and preventing the intra-mitochondrial accumulation of ATP, since elevated ATP levels lead to inhibition of the alternative oxidase, the only terminal oxidase in bloodstream from parasites, and consequently to reduced respiration [29]. Since functional mitochondria are essential for the maintenance of cellular integrity [30], destabilization and inhibition of ATP synthase would explain the observed ultrastructural effects.

In the case of *T. gondii*, the situation is less clear. The function of the protein encoded by ORF TgME49_319730 is unknown. The homolog with the highest similarity of which a structural model exists, the human mitochondrial import inner membrane translocase subunit Tim10, is part of the mitochondrial translocase Tim22 complex [25]. This complex has a size of approximately 440 kDa and consists of at least six different polypeptides, the protein Tim22 presumably forming a channel, and the Tim proteins 9, 10, and 29 and an acylglycerol kinase [31]. Mutations of proteins of this complex have been shown to impair protein import into mitochondria and lipid biosynthesis [32]. If the protein encoded by ORF TgME49_319730 is part of a similar complex in mitochondria of *T. gondii*, functional inhibition by drugs may result in a similar defect. This protein is certainly a candidate for further studies on mitochondria as a potential subcellular drug target.

## 4. Materials and Methods

### 4.1. Culture Media, Biochemicals and Compounds

Cell culture media was purchased from Gibco-BRL (Zürich, Switzerland), and biochemical agents were procured from Sigma, St. Louis, MO, USA). Compounds **1**, **2**, and **3** (see Figure 1) were synthesized as described in [Desiatkina et al., 2021, manuscript in prearation]. Compound **3** was purchased from TCI chemicals (Eschborn, Germany).

### 4.2. In Vitro Culture of Parasites

For electron microscopy, *T. gondii* ME49 tachyzoites were maintained and cultured in human foreskin fibroblasts as previously described [33]. For affinity chromatography necessitating high yields, tachyzoites were grown in Vero cells as previously described for *Neospora caninum* [34,35]. *T. brucei* TREU927 bloodstream forms were cultured as previously described [19]. The pellets of tachyzoites and bloodstream forms were stored at −80 °C.

### 4.3. In Vitro Proliferation/Viability Measurements

Inhibition of *T. gondii* tachyzoite proliferation and IC_50_ determination were performed using a transgenic *T. gondii* RH strain constitutively expressing β-galactosidase (*T. gondii* β-gal) as described for other trithiolato-bridged ruthenium–arene complexes [15,16]. Compounds were prepared as 1 mM stock solutions in DMSO (dimethyl sulfoxide, Sigma, St. Louis, MO, USA). For activity assays, HFF monolayers were cultured in 96-well plates by seeding 5 × 10^3^ HFF per well and allowing them to grow to confluence in phenol-red-free culture medium at 37 °C/5% CO_2_. For infection, *T. gondii* β-gal tachyzoites were separated from their host cells, parasites were isolated, and HFF monolayers were infected with freshly isolated tachyzoites (1 × 10^3^ per well), with compounds solutions added concomitantly during infection at 0.007, 0.01, 0.03, 0.06, 0.12, 0.25, 0.5, and 1 µM. After 72 h of culture, the medium was aspirated, and cells were permeabilized by adding 90 µL PBS containing 0.05% Triton X-100. After the addition of 10 µL 5 mM chlorophenol red-β-D-galactopyranoside (CPRG; Roche Diagnostics, Rotkreuz, Switzerland) dissolved in PBS, the absorption shift was measured at 570 nm wavelength at various time points using an EnSpire^®^ multimode plate reader (PerkinElmer, Inc., Waltham, MA, USA). The activity measured as the release of chlorophenol red over time was proportional to the number of live parasites down to 50 per well as determined in pilot assays. IC_50_ values were calculated after the logit-log-transformation of relative growth and subsequent regression analysis.

Screenings and IC_50_ calculations for *T. brucei* bloodstream forms were done as described previously [19] in 96-well flat-bottom plates, with each well containing 1.5 × 10^3^ parasites in 100 μL culture medium with or without a serial drug dilution. Ten 2-fold drug dilutions were used. After 72 h of compound exposure, 10 μL AlamarBlue^®^ (Resazurin; Sigma) was added to each well, allowing a color change via metabolic oxidation-reduction by viable trypanosomes during 2–3 h. Subsequently, plates were read with a Flexstation II microplate fluorimeter using an excitation wavelength of 536 nm and an emission wavelength of 588 nm. IC_50_ values were determined using the software GraphPad Prism 6.

Cytotoxicity assays using uninfected confluent HFF host cells were also performed by AlamarBlue assay. In brief, confluent HFF monolayers in 96-well plates were exposed to 0.1, 1, and 2.5 µM of each compound. Non-treated HFF as well as DMSO controls (0.01%, 0.1%, and 0.25%) were included. After 72 h of incubation at 37 °C/5% CO_2_, the medium was removed, and plates were washed once with PBS. Resazurin stock solution was diluted 1:200 in PBS, and 200 µL were added to each well. Plates were read at an excitation wavelength of 530 nm and emission wavelength of 590 nM by the EnSpire^®^ multimode plate reader (PerkinElmer, Inc.). Fluorescence was measured at different timepoints. Relative fluorescence units were calculated from timepoints with linear increase.

### 4.4. Transmission Electron Microscopy

Transmission electron microscopy (TEM) was performed as previously described [19,33,36]. In the case of *T. gondii*, confluent human foreskin fibroblasts grown in T25 flasks were infected with 10^6^
*T. gondii* ME49 tachyzoites and maintained at 37 °C/5% CO_2_ for 24 h. Subsequently, treatment with 500 nM of **1**, **2,** or **3**, or solvent controls were initiated. After 6, 12, 24, 48, and 72 h, the medium from the flasks was discarded; then, cells were washed in 0.1 M sodium cacodylate buffer (pH 7.3) and fixed in 2% glutaraldehyde in cacodylate buffer for 10 min at room temperature. The fixed monolayers were gently scraped from the flasks, transferred into Eppendorf tubes, and fixed for 2 h at room temperature. *T. brucei* bloodstream forms grown in 24-well tissue culture devices were treated with 200 nM of each compound in medium during 4 h. Fixation was done by washing cells once in ice-cold 0.1 M sodium cacodylate buffer, which was followed by fixation in 2% glutaraldehyde in cacodylate buffer for 2 h at room temperature. For both parasites, further preparation included post-fixation in 2% osmium tetroxide, pre-staining, stepwise dehydration in ethanol, and embedding in Epon-812 resin [19,33,36]. Following polymerization at 60 °C, ultrathin sections (80 nm) were cut using an ultramicrotome (Reichert and Jung, Vienna, Austria) and placed onto 200 mesh nickel grids (Plano GmbH, Marburg, Germany). Following staining, specimens were viewed on a Philips CM12 TEM operating at 80 kV.

### 4.5. Protein Extraction and Affinity Chromatography

For protein extraction, frozen pellets of *T. gondii* ME49 tachyzoites or *T. brucei* TREU927 bloodstream forms were resuspended in ice-cold extraction buffer, i.e., PBS (phosphate-buffered saline) containing 1% Triton X-100 and 1% of Halt proteinase inhibitor cocktail (ThermoFisher)). Suspensions were vortexed thoroughly and centrifuged (13,000 rpm, 10 min, 4 °C). Extraction of pellets was repeated twice. Three mL of extraction buffer was used in total. Supernatants were combined (resulting in approximately 3 mg of total protein) and subjected to affinity chromatography.

To produce the sepharose matrices conjugated to compounds 1, 2, and 3, 0.5 g of lyophilized epoxy-sepharose with a C12 spacer was suspended in 15 mL H_2_O and centrifuged at 300× *g* for 5 min. Washes in water were repeated twice followed by a wash with coupling buffer (0.1 M NaHCO_3_, pH 9.5). After the last wash, 20 mg of each compound dissolved in 2.5 mL DMSO (dimethylsulfoxide) were added, and coupling buffer was added to a maximum volume of 5 mL. Mock column medium was generated by incubating 0.5 g of epoxy-sepharose with DMSO. The mixture was incubated for 3 days at 37 °C under slow but continuous shaking to allow coupling of the epoxy group to the compounds. The resulting column medium (approximately 2 mL) was washed with coupling buffer (15 mL) followed by a wash with ethanolamine (1 M, pH 9.5) and by an incubation in 10 mL of ethanolamine for 4 h at 20 °C in the dark to block residual reactive groups. Then, the column medium was transferred to a chromatography column (Novagen, Merck, Darmstadt, Germany) and extensively washed with PBS-DMSO (1:1) and PBS to remove unbound compounds. The columns were stored in PBS containing 0.02% NaN_3_ at 4 °C.

Prior to affinity chromatography, mock columns were combined to either 1, 2, or 3 columns in tandem (mock first, then compound) and washed with 50 mL PBS equilibrated at 20 °C. Crude extracts (3 mL) prepared as described above were loaded onto the columns with a flow rate of 0.25 mL/min. The column was washed with PBS until the baseline was flat (10 column volumes, corresponding to 25 mL). Then, the columns were separated, and bound proteins were eluted with 50 mM acetic acid (5 mL per column). The eluates were lyophilized and stored at −80 °C.

### 4.6. Proteomic Analysis of the Eluted Proteins by Mass Spectrometry

The lyophilized eluates were dissolved in 10 μL of 8 M urea and 0.1 M of Tris-HCl^−^ (pH 8); then, 1 μL 0.1 M Tris-Cl^−^ (pH 8) buffer containing 0.1 M of dithiothreitol were added, which was followed by incubation for 30 min at 37 °C and constant mixing at 600 rpm. This step was repeated with 1 μL of 0.5 M of iodoacetamide. Iodoacetamide was quenched by the addition of 5 μL 0.1 M Tris-HCl^−^ (pH 8), and the urea concentration was further diluted to 4 M by the addition of 2 mM calcium dichloride in 20 mM Tris buffer. Proteins were digested for 2 h at 37 °C by the addition of 1 μL of 0.1 μg/μL LysC sequencing grade protease (Promega), which was followed by further dilution of urea to 1.6 M with the above calcium dichloride buffer and 1 μL of 0.1 μg/μL trypsin sequencing grade (Promega). Digestion was completed by incubation overnight at room temperature. Digestion was stopped by the addition of 2.5 μL 20% (*v*/*v*) trifluoroacetic acid. After an incubation for 15 min at room temperature, the digest was spun for 1 min at 16,000 g, and the cleared supernatant was transferred to a HPLC vial for subsequent nano-liquid reversed phase chromatography coupled to tandem mass spectrometry, as described elsewhere [37].

The mass spectrometry data were processed with MaxQuant (v1.6.14.0) against a current protein sequence databases from toxodb.org (ToxoDB-47_TgondiiME49) and tritrypdb.org (TriTrypDB-50_TbruceiTREU927) [34]. Then, the MaxQuant search results were processed in R-studio using an in-house R script exactly as described before [34].

Modeling of the structure of a binding protein was performed using the Swiss Model homology modeling tools [38] of the Swiss Model repository [39]. Protein–protein interaction networks were generated using the STRING knowledge base and software tool (https://string-db.org/; accessed on 12 February 2021).

### 4.7. Crude Membrane Fractions and Native Polyacrylamide Gel Electrophoresis (PAGE)

Crude mitochondrial membranes were obtained by digitonin extraction [40]. Briefly, 10^7^ trypanosomes were washed in TBS (10 mM Tris-HCl pH 7.5, 144 mM NaCl), suspended in 0.5 mL SoTE (20 mM Tris-HCl, pH 7.5, 0.6 M sorbitol, 0.2 mM EDTA) followed by the addition of 0.5 mL SoTE containing 0.05% (*w*/*v*) digitonin. After 5 min on ice, crude membranes were collected by centrifugation (6000× *g*, 5 min, 4 °C). For native PAGE, crude membranes were solubilized in 20 µL lysis buffer (20 mM Tris-HCl, pH 7.2, 15 mM KH_2_PO_4_, 20 mM MgSO_4_, 0.6 M sorbitol, 1.5% (*w*/*v*) digitonin), cleared, and supplemented with 10 times concentrated loading dye (100 mM Bis-Tris, pH 7, 500 mM aminocaproic acid, 5% (*w*/*v*) Coomassie Blue G). Proteins were separated on 3–12% NativePAGE gradient gels (Invitrogen Reinach, Switzerland) and transferred onto nitrocellulose membranes (Thermo Scientific, Waltham, MA, USA) using a semi-dry blotting system (BioRad, Cressier, Switzerland). Membranes were exposed to rabbit anti-ATP synthase subunit β or rabbit anti-TbAAC (kindly provided by Alena Ziková, Biology Centre of the Czech Academy of Sciences), diluted 1:1000 in TBS containing 5% (*w*/*v*) milk powder. Horseradish peroxidase-conjugated anti-rabbit (Dako, Glostrup, Denmark) was used at dilutions of 1:5000 and detected using an enhanced chemiluminescence detection kit (Thermo Scientific). Bands on blots were quantified using the gel analyzer function of Fiji [41].

## Figures and Tables

**Figure 1 ijms-22-10787-f001:**
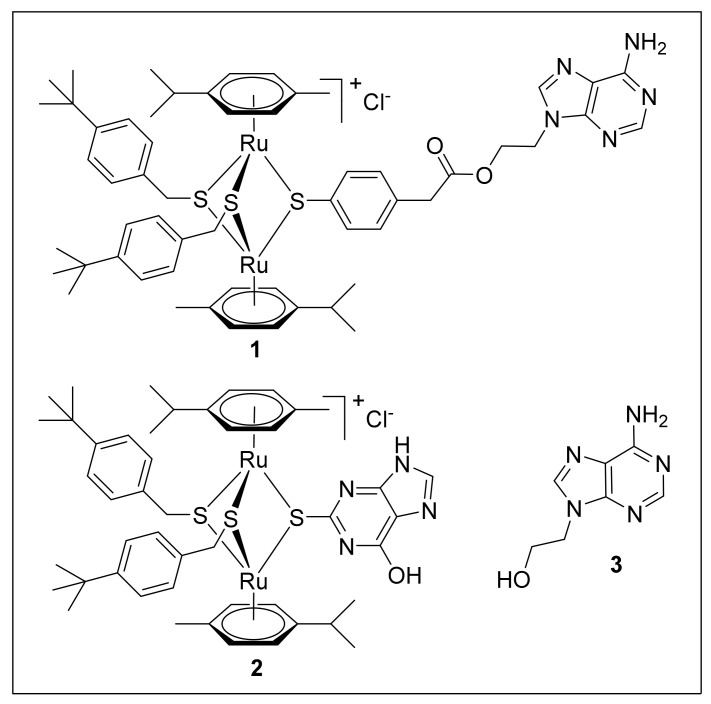
Structure of compounds **1**–**3** used in this study. **1** is a trithiolato–bridged dinuclear Ru(II) –arene conjugate containing a 9–(2–oxyethyl)–adenine unit, **2** is a trithiolato–bridged dinuclear Ru(II)–arene complex with 2–thioxanthine as one of the bridge thiols, and **3** is 9–(2–hydroxyethyl) –adenine.

**Figure 2 ijms-22-10787-f002:**
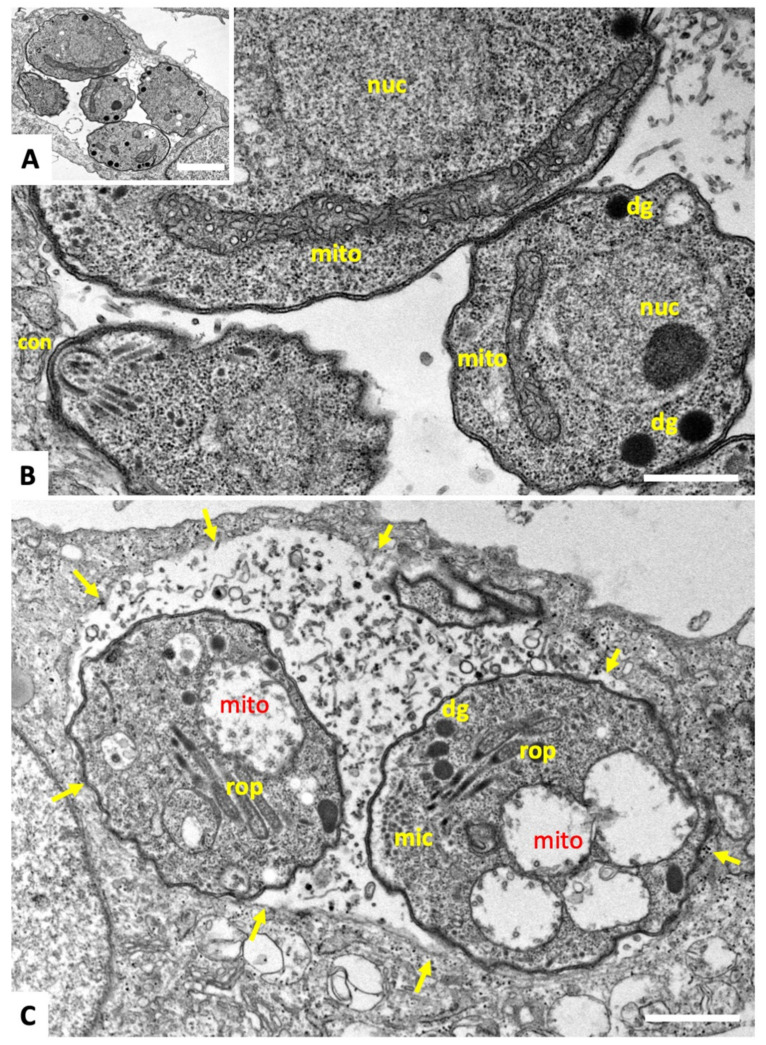
TEM of *T. gondii* tachyzoites grown in human foreskin fibroblasts. (**A**,**B**) are non-treated tachyzoites, (**C**) shows tachyzoites treated with 500 nM of compound **1** for 6 h. **B** is a higher magnification view of **A**. The arrows delineate the membrane of the parasitophorous vacuole; nuc = nucleus, rop = rhoptries, dg = dense granules, mic = micronemes, mito = mitochondrion, con = conoid. Bar in **A** = 1.0 µm; in **B** and **C** = 0.3 µm.

**Figure 3 ijms-22-10787-f003:**
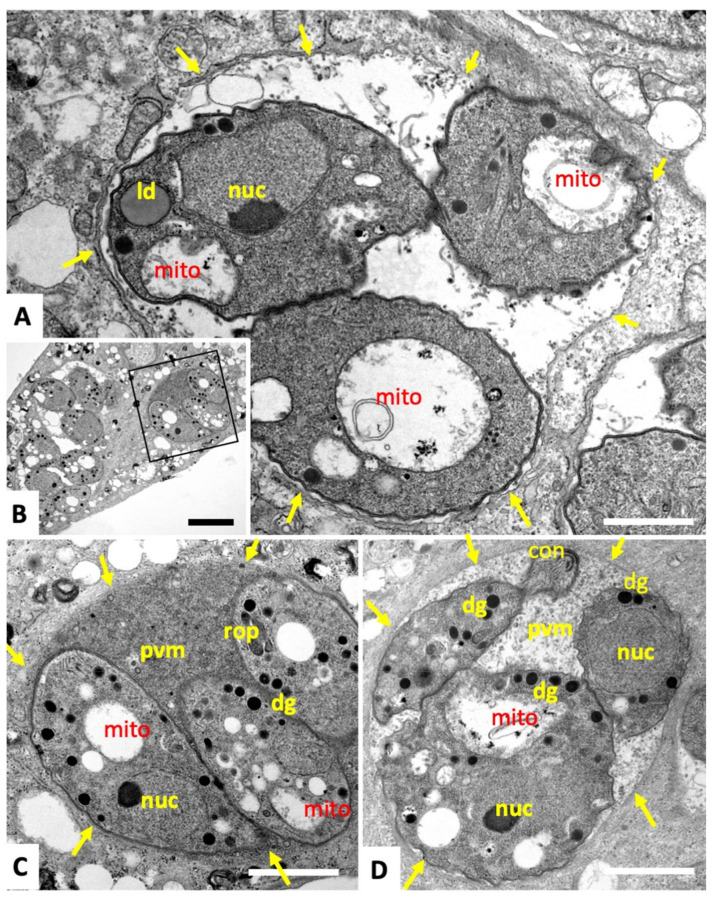
TEM of *T. gondii* tachyzoites grown in human foreskin fibroblasts and treated with 500 nM of compound **1**. Panel (**A**) shows parasites treated for 6 h, exhibiting a profoundly altered mitochondrion. (**B**,**C**) show infected cultures treated for 12 h, with C being a higher magnification view of the boxed area in **B**. (**D**) shows a culture maintained in the presence of **1** for 24 h. The arrows delineate the membrane of the parasitophorous vacuole; ld = lipid droplet, nuc = nucleus, rop = rhoptries, dg = dense granules, mic = micronemes, mito = mitochondrion, con = conoid, pvm = parasitophorous vacuole matrix. Bar in **A** = 0.5 µm; in **B** = 2.2 µm; in **C** and **D** = 1.1 µm.

**Figure 4 ijms-22-10787-f004:**
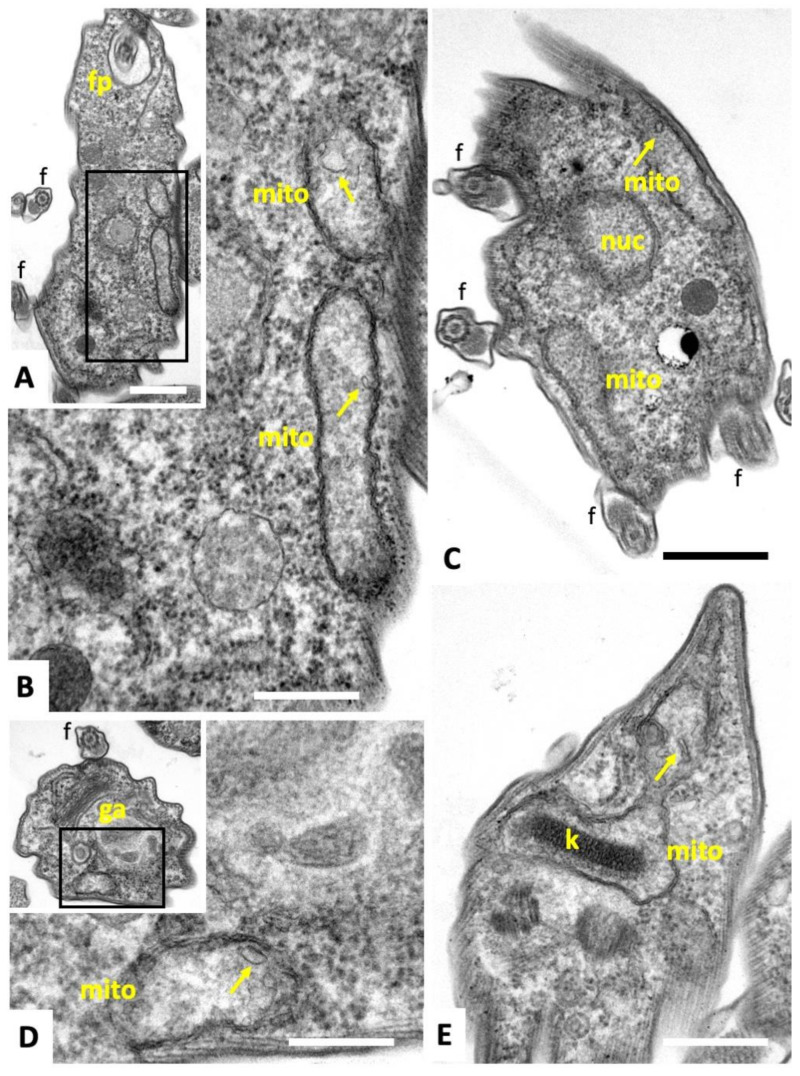
TEM of *T. brucei* bloodstream forms. (**A**,**B**) shows longitudinal sections through the cytoplasm of a parasite cultured in the absence of compounds, with the boxed area in **A** magnified in **B** highlighting parts of the mitochondrion. (**C**,**D**) show cross-sections of parasites cultured in the presence of compounds **2** and **3**, respectively, both at 200 nM for 4 h. In **D**, the boxed area in the low magnification view is enlarged. (**E**) shows a section through the posterior part of a parasite treated with **3**. Note the highly ordered kinteoplast structure (k) and partially exposed mitochondrion (mito) that exhibits a rather amorphous and slightly electron-dense matrix with only few cristae-like structures (arrows); ga = golgi apparatus, k = kinetoplast, f = flagellum, fp = flagellar pocket. Bar in **A** = 0.35 µm; in **B** = 0.2 µm; in **C** = 0.3 µm; in **D** = 0.2 µm; in **E** = 0.3 µm.

**Figure 5 ijms-22-10787-f005:**
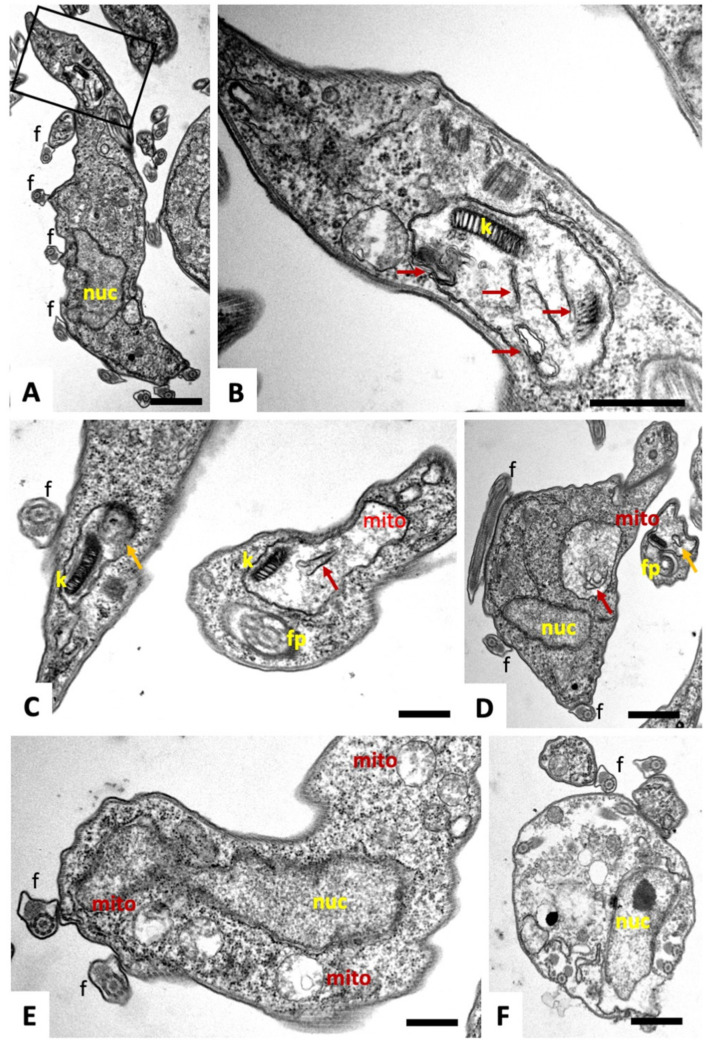
TEM of *T. brucei* bloodstream forms treated with 200 nM of compound **1** for 4 h. (**A**) is a lower magnification view of a longitudinal section, the boxed region at the posterior end is magnified in (**B**). (**C**) shows two other parasites with altered kinetoplast (k), (**D**,**E**) show sections through the mitochondrion in other parts of the cell. Arrows point to the structurally altered mitochondrial matrix, characterized by low electron density and potentially membranous content of unknown nature (red arrows), and membrane fragments (orange arrows). (**F**) shows parasites with a more severely altered structural organization and apparent loss of cytoplasmic content. mito = mitochondrion, k = kinetoplast, f = flagellum, fp = flagellar pocket, nuc = nucleus. Bars in **A** = 0.8 µm, **B** = 0.35, **C** = 0.21 µm, **D** = 0.4 µm, **E** = 0.21 µm, **F** = 0.35 µm.

**Figure 6 ijms-22-10787-f006:**
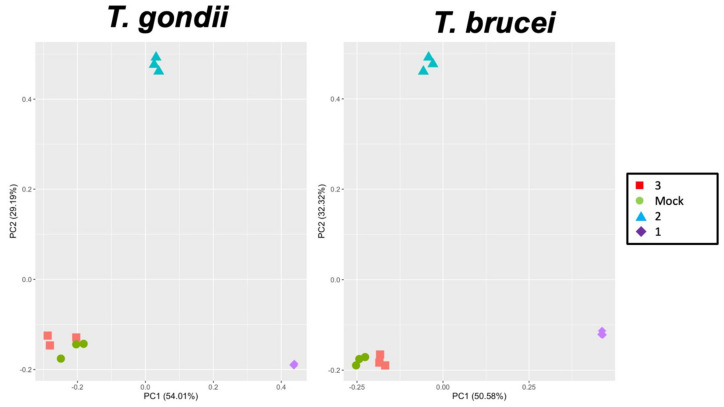
Principal component analysis of proteome dataset from differential pull–downs. Pull–down fractions of cell–free extracts from *T. gondii* and *T. brucei* from mock, **3**, **2,** or **1** columns were compared by MS shotgun analysis as described in Materials and Methods. The LFQ values of technical replicates are shown.

**Figure 7 ijms-22-10787-f007:**
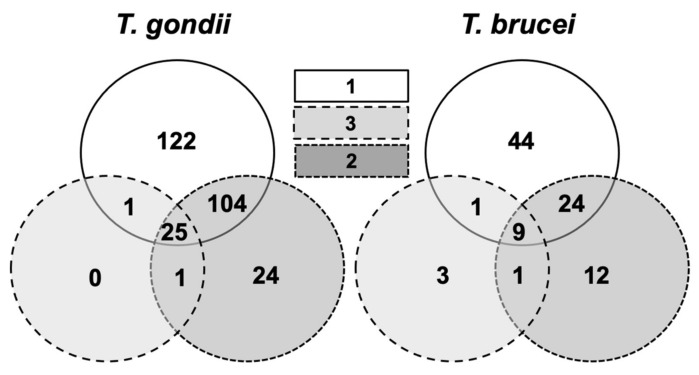
Venn diagram detailing the number of proteins from differential pull-downs. Pull-down fractions of extracts from *T. gondii* and *T. brucei* from **1, 2,** and **3** columns were compared by MS shotgun analysis as described in Materials and Methods.

**Figure 8 ijms-22-10787-f008:**
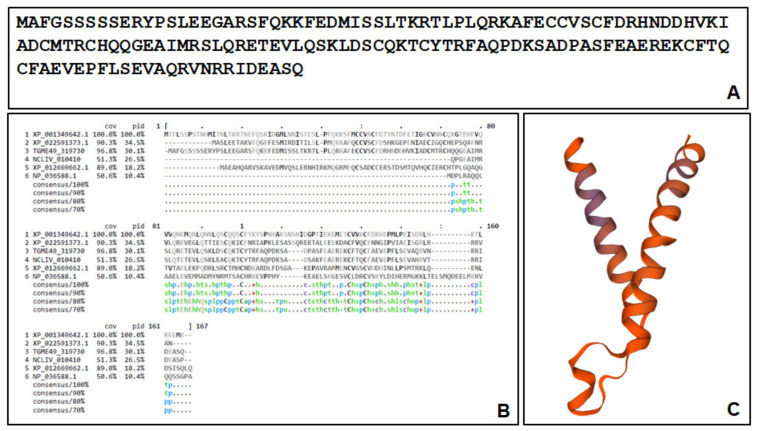
Deduced primary sequence of ORF TgME49_319730 is shown in (**A**). Alignments to five homologs and consensus sequences are presented in (**B**). Amino acid sequences are shown in grey, black amino acids indicate homologies or identities, from the consensus sequences, s, h and l are indicated in green, p in blue, (**C**) Model structure calculated from the mitochondrial import membrane translocase subunit Tim10 encoded by NP_036588.1 with hydrophilic residues depicted in violet and hydrophobic residues in orange.

**Figure 9 ijms-22-10787-f009:**
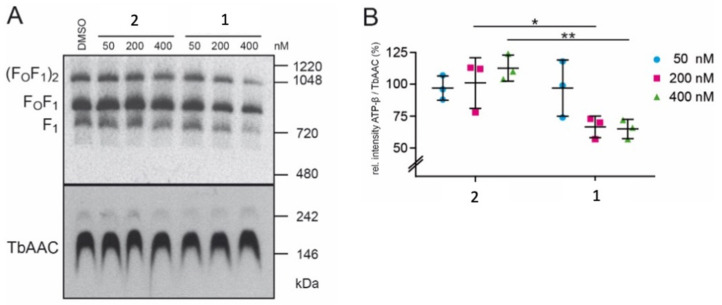
Native PAGE analysis and quantification of ATP-synthase complexes. (**A**) shows native PAGE and immunoblot analyses of mitochondria-enriched membranes after treatment with DMSO (mock control), or compound **1** or **2** for 1 h. ATP synthase F_1_ and F_o_F_1_ monomers and dimers ((F_o_F_1_)_2_) were visualized using the anti-ATP synthase subunit β antibody (ATP-β), the loading control TbAAC was visualized with anti-TbAAC antibody. (**B**) shows protein quantification of samples from A analyzed by SDS-PAGE and immunoblotting. Ratios of signal intensities of ATP-β versus TbAAC were compared to DMSO-treated samples (n ≥ 3). * *p* < 0.05, ** *p* < 0.01.

**Table 1 ijms-22-10787-t001:** IC_50_ values of the three compounds used in this study.

Compound	MW	IC_50_ *T. gondii* (nM) *	IC_50_ *T. brucei* (nM)
**1**	1193	59	29
**2**	1032	>1000	>1000
**3**	179	>1000	>1000

* values presented in Desiatkina et al., 2021, manuscript in preparation.

**Table 2 ijms-22-10787-t002:** List of the 20 most abundant proteins from *T. gondii* ME49 specifically binding to the compound **1** column, as identified by differential affinity chromatography followed by mass spectrometry. See Appendix A for the full dataset. The abundance is given as the percentage of the iBAQ values, with 100% being the sum of the iBAQ values of all 122 proteins specifically binding to **1**. Mean values ± standard errors of three replicates are given.

ORF Number	Annotation	iBAQ (% of Total)
TGME49_319730	YOU2 family C2C2 zinc finger protein	30.3 ± 1.5
TGME49_316710	Hypothetical protein	7.1 ± 1.1
TGME49_278270	Nucleolar protein, structural component of H/ACA snoRNPs, putative	5.1 ± 1.0
TGME49_215350	Hypothetical protein	4.6 ± 0.2
TGME49_263990	Hypothetical protein	4.3 ± 0.3
TGME49_214940	MIC2-associated protein M2AP	3.8 ± 0.3
TGME49_221510	Hypothetical protein	2.6 ± 0.3
TGME49_236040	Fructose-1,6-bisphosphate aldolase	2.5 ± 0.3
TGME49_221620	Beta-tubulin, putative	1.9 ± 0.5
TGME49_264040	Hypothetical protein	1.9 ± 0.1
TGME49_288245	Hypothetical protein	1.8 ± 0.1
TGME49_215430	Hypothetical protein	1.6 ± 0.3
TGME49_294670	Translation initiation factor 3 subunit	1.5 ± 0.1
TGME49_226410	EF-1 guanine nucleotide exchange domain-containing protein	1.3 ± 0.3
TGME49_218410	Ribosomal protein RPP0	1.1 ± 0.2
TGME49_245620	Ribosomal-ubiquitin protein RPS27A	1.0 ± 0.2
TGME49_247770	Hypothetical protein	0.9 ± 0.1
TGME49_289830	Eukaryotic initiation factor-3, delta subunit, putative	0.9 ± 0.1
TGME49_287210	Proteasome subunit alpha2, protease of the acylase family	0.8 ± 0.1
TGME49_203630	Ribosomal protein RPL44	0.8 ± 0.1

**Table 3 ijms-22-10787-t003:** List of the 20 most abundant proteins from *T. brucei* TREU927 specifically binding to a complex **1** column, as identified by differential affinity chromatography followed by mass spectrometry. See Appendix A for the full dataset. The abundance is given as the percentage of the iBAQ values, with 100% being the sum of the iBAQ values of the 48 proteins specifically binding to **1**. Mean values ± standard errors of three replicates are given.

ORF Number	Annotation	iBAQ (% of Total)
Tb927.5.3090	Mitochondrial ATP synthase subunit, putative	24.9 ± 4.6
Tb927.10.5050	Mitochondrial ATP synthase epsilon chain	20.0 ± 4.2
Tb927.5.2930	Mitochondrial ATP synthase subunit, putative	13.3 ± 1.5
Tb927.5.1160	Degradation arginine-rich protein for mis-folding, putative	4.1 ± 0.6
Tb927.11.600	Mitochondrial ATP synthase subunit, putative	3.4 ± 0.5
Tb927.3.3330	Heat shock protein 20, putative	2.4 ± 0.2
Tb927.4.4910.1	3,2-trans-enoyl-CoA isomerase, mitochondrial precursor, putative	2.3 ± 0.3
Tb927.3.2880	Mitochondrial ATP synthase subunit, putative	2.3 ± 0.3
Tb927.3.3750	Paraflagellar rod component, putative	2.0 ± 0.0
Tb927.10.13110	Outer arm dynein light chain 7	2.0 ± 0.6
Tb927.10.4310	Prohibitin 2, putative	1.6 ± 0.1
Tb927.4.3590	Translation elongation factor 1-beta, putative	1.6 ± 0.3
Tb927.7.2780	Hypothetical protein, conserved	1.5 ± 0.7
Tb927.6.4250	Hypothetical protein, conserved	1.4 ± 0.2
Tb927.9.4680	Eukaryotic initiation factor 4A-1	1.4 ± 0.1
Tb927.6.3290	Intraflagellar transport protein 20	1.2 ± 0.3
Tb927.8.4810	Prohibitin 1	1.1 ± 0.1
Tb927.10.2080	Hypothetical protein, conserved	1.1 ± 0.4
Tb927.5.4500	Ras-like small GTPase, putative	1.0 ± 0.1
Tb927.3.5550	Intraflagellar transport protein 27	1.0 ± 0.2

## Data Availability

Data are made available as supplementary datasets (see above).

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
