# Peer review of "Cellular and Molecular Targets of Nucleotide-Tagged Trithiolato-Bridged Arene Ruthenium Complexes in the Protozoan Parasites Toxoplasma gondii and Trypanosoma brucei"

_ijms, 2021, doi:10.3390/ijms221910787_

Round 1
Reviewer 1 Report
In this manuscript, the authors Anghel et al. assessed a trithiolato-bridged ruthenium complex for anti-parasitic activity in the diverse microbes Toxoplasma gondii and Trypanosoma brucei. Two additional compounds were used as controls. Compound 1 was shown to inhibit proliferation of T. gondii and T. brucei and, using electron microscopy, morphological defects in T. brucei and possibly T. gondii mitochondria were observed. Compound-binding proteins (for all three compounds) in both parasites were identified using pull-down assays.
Major points:
- The authors did not address whether the compounds had any effects on the host mammalian cells (HFF). Do these compounds affect host cell growth? Are there any morphological changes in the host cells?
- The connection between compound 1 and the gondii mitochondria is less conclusive. It is unclear if the parasite organelle affected by compound 1 is mitochondria from the TEM sections shown. Plus, the homology between TGME49_319730 (the most abundant compound 1 specific protein identified) and human Tim10 seems low (12.5% identity). Combined with the fact that no mitochondrial ATP-synthase subunits but vacuolar ones were identified in T. gondii suggest that perhaps a vacuolar compartment is disrupted in Toxoplasma.
- Using immuno-EM with an antibody against a mitochondrial marker would help show that this compartment is indeed the mitochondria. Alternatively, immunofluorescence microscopy (using a mitochondrial marker) may be used to detect any abnormalities in the parasite mitochondria albeit with less resolution.
Minor points:
- The method for the viability/proliferation assays could be more clearly described instead of just listing a citation.
Author Response
Many thanks for the valuable input. Here are our responses to the comments
The authors did not address whether the compounds had any effects on the host mammalian cells (HFF). Do these compounds affect host cell growth? Are there any morphological changes in the host cells?
Response: The compounds do not have any notable effects on the host mammalian cell viability at concentrations up to 2.5 µM, and also no morphological changes were noted. This information was introduced into the results section (lines 126-127)
- The connection between compound 1 and the gondii mitochondria is less conclusive. It is unclear if the parasite organelle affected by compound 1 is mitochondria from the TEM sections shown. Plus, the homology between TGME49_319730 (the most abundant compound 1 specific protein identified) and human Tim10 seems low (12.5% identity). Combined with the fact that no mitochondrial ATP-synthase subunits but vacuolar ones were identified in T. gondii suggest that perhaps a vacuolar compartment is disrupted in Toxoplasma.
- Using immuno-EM with an antibody against a mitochondrial marker would help show that this compartment is indeed the mitochondria. Alternatively, immunofluorescence microscopy (using a mitochondrial marker) may be used to detect any abnormalities in the parasite mitochondria albeit with less resolution.
Response: We do not agree with the statement that the connection between compound 1 and the T. gondii mitochondrion is not conclusive, for the following reasons:
- First, we have shown earlier that ruthenium-based organometallic complexes target the mitochondrion in gondii (Basto et al., 2017, Desiatkina et al., 2021) and in the closely related Neospora caninum (Basto et al., 2019), as well as in T. brucei (Jelk et al., 2019).
- Secondly, the mitochondrion in gondii is a structural entity which is easily identified in TEM sections. It has an electron-dense matrix with a distinct cristae-like membranous pattern, as shown in Figure 2A and described in lines 138-140). These structural entities are not present anymore in compound 1 treated parasites. Instead they contain large vacuoles, in which residues of the mitochondrial matrix are partially still visible (Fig. 2 and 3).
- In addition, these alterations in the mitochondrial matrix have been described in multiple papers describing ultrastructural changes induced by drugs specifically acting on mitochondria such as endochin-like quinolones, buparvaquone and decoquinate in gondii and related apicomplexan parasites. We thus do not think that performing any kind of labeling will add anything meaningful to the message of this paper.
- With respect to the major compound 1 binding protein: if human TIM10 would be a homolog with a high identity, one would expect that it would also bind to, and also be affected, by compound 1, but since compound 1 does not affect human fibroblasts, we cannot expect a high degree of identity (with 12.5% identify being substantial already). We do not claim that gondii TIM10 homologue is the actual target, but indicate that this protein, annotated as a YOU2 family C2C2 zinc finger protein, is wrongly annotated, and could potentially be a TIM10 homologue, with more studies required to confirm this aspect.
However, the reviewer is of course right, that many other targets could be involved and contribute to the activity of compound 1, some of which might fulfill important metabolic functions in other cellular compartments. We believe that we make this also evident in the discussion, where examining the protein-protein-interaction network generated by the STRING knowledge base and software tool (Swiss Institute of Bioinformatics) for the compound 1-binding proteins from T. gondii (Fig. S1) reveals also an involvement of the translation machinery.
Minor points:
- The method for the viability/proliferation assays could be more clearly described instead of just listing a citation.
We added a section describing the viability/proliferation assays
Reviewer 2 Report
The manuscript under analysis crowns the work of the team, paving the way for the applications of ruthenium complexes as antiparasitic chemical species, active on two of the most pathogenic monocellular eukaryotes.
Authors exploit a vulnerability of the said parasites: the need to import essential metabolites, such as purine nitrogenous bases.
In an ingenious and systematic manner, the authors identify the target proteins of the ruthenium complex decorated with adenine derivatives. Most of these proteins play essential metabolic roles in parasite cells, at the level of mitochondria and of protein biosynthesis machinery. In addition, the paper investigates (by TEM technique) the intracellular locations of the complexes used, as well as the effect of fixing the complexes to the components of cellular organelles.
I have no criticism on the manuscript text, but a broader explanation of the reasons for choosing the structure of ruthenium complexes would be useful.
There are very few inaccuracies in the text, such as the non-use of italic fonts for the names of biological species.
Author Response
Thank you for your valuable comments, which we of course appreciate.
I have no criticism on the manuscript text, but a broader explanation of the reasons for choosing the structure of ruthenium complexes would be useful.
Response: Actually, the reasons for choosing the ruthenium complexes used in this study are outlined in the introduction (lines 90-98). These compounds were chosen based on the insights that many parasites lack de novo synthesis pathways of essential metabolites and have therefore become auxotrophic. We have previously found that organometallic ruthenium complexes exhibit interesting activities against protozoan parasites, and we hypothesized that conjugating Ru(II)-complexes with metabolites these parasites need to scavenge from the host could improve their uptake and binding to specific targets. Purine auxotrophy is a classical features of many protozoans, and we have designed various conjugates based on a trithiolato-bridged dinuclear ruthenium(II)-arene scaffold with anchored purines or purine analogues [Desiatkina et al., 2021, manuscript in preparation]. Compounds 1 and 2 were chosen for these studies.
There are very few inaccuracies in the text, such as the non-use of italic fonts for the names of biological species.
Response: We re-read the manuscript an eliminated typos and inaccuracies, hopefully completely
Round 2
Reviewer 1 Report
The changes are acceptable